# Effect of *Lacticaseibacillus paracasei* Strain Shirota on Improvement in Depressive Symptoms, and Its Association with Abundance of Actinobacteria in Gut Microbiota

**DOI:** 10.3390/microorganisms9051026

**Published:** 2021-05-10

**Authors:** Machiko Otaka, Hiroko Kikuchi-Hayakawa, Jun Ogura, Hiroshi Ishikawa, Yukihito Yomogida, Miho Ota, Shinsuke Hidese, Ikki Ishida, Masanori Aida, Kazunori Matsuda, Mitsuhisa Kawai, Sumiko Yoshida, Hiroshi Kunugi

**Affiliations:** 1National Centre of Neurology and Psychiatry, Department of Mental Disorder Research, National Institute of Neuroscience, 4-1-1 Ogawa-Higashi, Kodaira-shi, Tokyo 187-8502, Japan; otaka.mc@gmail.com (M.O.); Ogura-Jun@ncnp.go.jp (J.O.); yomogida@ncnp.go.jp (Y.Y.); ota@ncnp.go.jp (M.O.); shidese@ncnp.go.jp (S.H.); iishida@ncnp.go.jp (I.I.); 2Yakult Central Institute, 5-11 Izumi, Kunitachi-shi, Tokyo 186-8650, Japan; hiroko-hayakawa@yakult.co.jp (H.K.-H.); h-ishikawa@yakult.co.jp (H.I.); masanori-aida@yakult.co.jp (M.A.); kazunori-matsuda@yakult.co.jp (K.M.); mitsuhisa-kawai@yakult.co.jp (M.K.); 3National Centre of Neurology and Psychiatry, Department of Psychiatry, 4-1-1 Ogawa-Higashi, Kodaira-shi, Tokyo 187-8551, Japan; syoshida@ncnp.go.jp; 4Department of Psychiatry, Teikyo University School of Medicine, 2-11-1 Kaga, Itabashi-ku, Tokyo 173-8605, Japan

**Keywords:** probiotic, major depressive disorder, bipolar disorder, gut microbiota, Actinobacteria

## Abstract

We previously reported lower counts of lactobacilli and Bifidobacterium in the gut microbiota of patients with major depressive disorder (MDD), compared with healthy controls. This prompted us to investigate the possible efficacy of a probiotic strain, *Lacticaseibacillus paracasei* strain Shirota (LcS; basonym, *Lactobacillus casei* strain Shirota; daily intake of 8.0 × 10^10^ colony-forming units), in alleviating depressive symptoms. A single-arm trial was conducted on 18 eligible patients with MDD or bipolar disorder (BD) (14 females and 4 males; 15 MDD and 3 BD), assessing changes in psychiatric symptoms, the gut microbiota, and biological markers for intestinal permeability and inflammation, over a 12-week intervention period. Depression severity, evaluated by the Hamilton Depression Rating Scale, was significantly alleviated after LcS treatment. The intervention-associated reduction of depressive symptoms was associated with the gut microbiota, and more pronounced when Bifidobacterium and the Atopobium clusters of the Actinobacteria phylum were maintained at higher counts. No significant changes were observed in the intestinal permeability or inflammation markers. Although it was difficult to estimate the extent of the effect of LcS treatment alone, the results indicated that it was beneficial to alleviate depressive symptoms, partly through its association with abundance of Actinobacteria in the gut microbiota.

## 1. Introduction

Mood disorders, including major depressive disorder (MDD) and bipolar disorder (BD), are a leading cause of disability worldwide, affecting over 300 million people [1]. Both disorders present with recurrent depressive episodes, while manic symptoms also manifest in BD. Although the pathophysiology of mood disorders remains elusive, increasing attention has been paid to an association with dysbiosis of the gut microbiota [2]. Accumulating observations have revealed that the gut microbiota influence the brain functions and psychological state of their host via the gut–brain axis [3]. The gut–brain axis refers to bidirectional communication between the brain and the gut. Signals from the brain modify the motility, sensory, and secretory modalities of the gastrointestinal tract and, in turn, signals from the gut can affect emotional behavior and stress- and pain-modulation systems [4]. An early observation that, under restraint stress, germ-free (GF) mice showed enhanced secretion of stress markers (plasma adrenocorticotropic hormone and corticosterone), when compared with specific-pathogen-free mice, indicated that the gut microbiota affect host response to stress [5]. A study of fecal microbiota transplantation from depressive patients into GF mice showed that the recipient mice displayed more depression-like behavior than control mice [6]. It has been suggested that microbial cellular components and metabolites of the complex gut microbiota may influence brain functions via blood circulation, humoral pathways, and the immune system, as well as via neuronal pathways [4,7,8,9,10].

Alterations in the gut microbiota composition have been characterized in several human studies [11,12,13,14,15,16]. We previously reported lower counts of lactobacilli and Bifidobacterium in the gut microbiota of patients with MDD, compared with healthy controls [17]. In patients with BD, in contrast, we did not find such a difference; however, we observed a negative correlation between Bifidobacterium counts and plasma cortisol levels, and between lactobacilli counts and sleep disturbance, in the patients [18]. These observations were supported by a number of studies in animal models. Reduced intestinal lactobacilli levels were associated with the development of stress-induced despair behavior in mice, which was improved by supplementation of *Limosilactobacillus reuteri* (formerly referred to as *Lactobacillus reuteri*, before a taxonomic reclassification of the Lactobacillus genus [19]) [20]. Another study showed that microbiota transplantation transmitted depressive behavioral symptoms and reduced endocannabinoid (eCB) signaling (due to lower peripheral levels of fatty acid precursors of eCB ligands), and which were also alleviated by complementation with a lactobacilli strain [21]. Maternal separation caused disruption of the gut microbiota, and a significant decrease in lactobacilli, in infant rhesus monkeys [22]; and the exaggerated stress response of the hypothalamic–pituitary–adrenal axis in GF mice was suppressed by reconstitution with *Bifidobacterium infantis* [5]. In line with the idea of maintaining psychological homeostasis via the gut–brain axis, several human trials have been performed to evaluate the function of probiotics containing lactobacilli and bifidobacterial strains, in controlling anxiety and depression [23,24]. However, there remains limited evidence for the efficacy of probiotics in treating mood disorders.

To evaluate the possible efficacy of probiotic treatment in alleviating depressive symptoms, we conducted a single-arm trial, using *Lacticaseibacillus paracasei* strain Shirota (LcS; formerly referred to as *Lactobacillus casei* strain Shirota), in patients with MDD or BD. LcS is a widely used probiotic strain, and a number of clinical studies have shown its physiologically beneficial functions involving normalization of altered gut microbiota [25,26], immune control [27,28,29,30], and modulation of psychological homeostasis via the gut–brain axis [31,32]. We assessed changes in psychiatric symptoms with LcS treatment for 12 weeks; and also analyzed the gut microbiota composition and biological markers for intestinal permeability and inflammation, in a search for variables related to treatment outcomes. The study will provide key new findings that will help elucidate the mechanisms of the psychological effects of probiotics.

## 2. Materials and Methods

### 2.1. Participants

Participants were patients with MDD or BD aged between 20 and 65 years. Diagnosis was made by a board-certified research psychiatrist, according to the Diagnostic and Statistical Manual of Mental Disorders, 5th Edition (DSM-5) [33]. The majority of the participants were receiving psychotropic medications; however, the medications at baseline remained unchanged over the intervention period. The patients were rated for their depressive symptoms using the Hamilton Depression Rating Scale, 21-item version (HAM-D21) [34]. To see the effect of LcS in patients with a wide range of severity, those whose HAM-D21 score was 11 or more were included in the study. One individual was excluded because of strong suicidal ideation (score for the item “Suicide” in HAM-D21 was 3 or more). Information on years of education and Body Mass Index (BMI; kg/m^2^) was also collected. A generalized propensity to be anxious was evaluated by a self-report survey using the trait anxiety scale of the State-Trait Anxiety Inventory (STAI) [35], and the score was interpreted and classified into five categories [36].

### 2.2. Study Product

Fermented milk containing at least 4.0 × 10^10^ colony-forming units (CFUs) of *Lacticaseibacillus paracasei* strain Shirota YIT 9029 (LcS) per bottle (80 mL) was used as the study product. LcS has a long history of safe use as a food material over 80 years, and the United States Food and Drug Administration has accredited it as Generally Recognized As Safe (GRAS) [37]. The strain was obtained from the Culture Collection Research Laboratory of the Yakult Central Institute (Kunitachi-shi, Tokyo, Japan). The study product was distributed and stored below 10 °C.

### 2.3. Study Design

The study was conducted in Tokyo, Japan, from January 2018 to March 2020, as an interventional study with a before-and-after design. The study protocol was registered at the UMIN (University Hospital Medical Information Network) Clinical Trials Registry (ID: UMIN000032825). The study consisted of a screening period, followed by a pre-intervention period and 12-week intervention period. All the patients consumed a daily dose of two bottles of the study product (total daily dose: 160 mL containing at least 8.0 × 10^10^ CFUs of LcS) throughout the intervention period. During the study, psychiatric evaluations were performed on the patients at baseline 0, 6, and 12 weeks; and patients submitted stool samples for the gut microbiota analysis at the same time points.

### 2.4. Psychiatric Evaluation

The patients were rated for their depressive symptoms using the HAM-D21. The Beck Depression Inventory (BDI) [33,38] was used for self-report assessments of depressive symptoms. The Pittsburgh Sleep Quality Index (PSQI) [39] was used for retrospective self-assessment of sleep quality and sleep disturbance over a one-month period, and the state anxiety scale of the STAI for evaluating anxiety levels at the time of answering. The gastrointestinal (GI) symptoms were assessed using the Gastrointestinal Symptom Rating Scale (GSRS), and the presence or absence of irritable bowel syndrome (IBS) or functional bowel disorders (FBD) was assessed using the ROME IV criteria [40].

### 2.5. Gut Microbiota Analysis

The gut microbiota analysis was performed using the method described by Matsuda et al. [41] (known as the Yakult-Intestinal-Flora-Scan: YIF-SCAN [42]). A portion of defecated stool was collected in a plastic tube containing the RNA stabilization solution, Ambion RNA*later* (Thermo Fisher Scientific; Waltham, MA, USA). Total RNA was extracted from each stool sample and subjected to reverse transcription-quantitative PCR (RT-qPCR) targeting bacterial rRNA molecules. The specific primer sets were used for the RT-qPCR assay to measure the bacterial counts (log_10_ cells/g of stool) of the 19 bacterial groups or species in Appendix A. The bacterial count of total lactobacilli was calculated from the sum of the counts of the 8 subpopulations. The lower detection limit of the assay varied between the targets but fell within the range of 2.0 to 5.0 log_10_ cells/g of stool. Bacterial counts of “not detected” samples were regarded as representing half the detection limit in the statistical analysis.

### 2.6. Biological Markers for Intestinal Permeability and Inflammation

Intestinal permeability was measured using the Intestinal Permeability Assessment (Genova Diagnostics; Asheville, NC, USA) according to the manufacturer’s instructions. The patients consumed a drink containing a premeasured amount of lactulose and mannitol, and then collected their urine over the next six hours. The urinary excretion ratio of each of the ingested sugars was measured, and the lactulose-to-mannitol excretion ratio (L/M ratio) was calculated as the degree of intestinal permeability. The level of high-sensitivity C-reactive protein (hs-CRP) in the blood was measured at SRL Inc. (Shinjuku-ku, Tokyo, Japan), using serum fractions separated after centrifugation at 3000 rpm for 10 min. The urine and blood samples were collected from each patient after 12 h of overnight fasting.

### 2.7. Statistical Analysis

The statistical analysis was performed using R v. 3.5.3 software (R Foundation for Statistical Computing, Vienna, Austria). The PMCMRplus package was used to assess statistical differences in the data between the three time points. The multiple comparison was performed using the Friedman test, and when significance was observed, the Nemenyi test was conducted post hoc for pairwise comparisons with the baseline data (0 weeks). McNemar’s Chi-squared test was employed to compare paired proportions between the time points using the exact 2 × 2 package in R, where categories with no observations in all the time points were excluded from the analysis. In the responder analysis, Wilcoxon’s rank sum test and Fisher’s exact test for 2 × 3 and 2 × 2 contingency tables were used for the inter-group comparison of ratio and nominal data, respectively. Kendall’s rank correlation coefficient (*τ*) was calculated to assess statistical associations between the bacterial counts and change in the HAM-D21 score during the study. The glm R package was used for multiple linear regression analysis. The linear regression analysis used the change in the HAM-D21 score over the study as a response variable; and gender, age, disease (MDD/BD), baseline score in the HAM-D21, and the data of each bacterial target as explanatory variables. The fixed effect size of the explanatory variable, and its 95% confidence interval (CI) and *p*-value: Pr(>|t|), were estimated. All statistical tests were performed in a two-tailed test at the 0.050 level of significance. All *p*-values were rounded to three decimal places and are presented as “*p* < 0.001” if they were below 0.001 after rounding.

## 3. Results

### 3.1. Subject Disposition

Initially, a total of 22 subjects were enrolled. However, two subjects SN6 and SN16 were later excluded due to withdrawal of the informed consent during the study. Of the 20 subjects who completed the study, two who violated the protocol (SN22 used antibiotics for *Helicobacter pylori* eradication, and SN19 did not submit stool samples) were excluded, leaving 18 subjects for the analysis. The characteristics of the 18 subjects (14 females and 4 males; 15 MDD and 3 BD) are shown in Table 1. Among them, 15 were receiving psychotropic medications, with a mean daily antipsychotic dose of 117.1 mg (chlorpromazine equivalent) and antidepressant dose of 113.3 mg (imipramine equivalent), according to the published guidelines [43]. Six received one or two mood stabilizers (valproic acid, lithium, or lamotrigine). The distribution of the subjects, based on the trait anxiety scale scores of the STAI, was as follows: very low (0), low (0), normal (0), high (4), and very high (14). No adverse events attributable to the intervention were observed during the study period.

### 3.2. Changes in Rating Scales of Depressive, Sleep, and Gastrointestinal Symptoms

The HAM-D21 total score decreased significantly over the 12-week intervention period (mean ± SD: 0 weeks 17.7 ± 4.1, 12 weeks 10.9 ± 7.3; *p* < 0.001; (Figure 1a). The BDI score tended to decrease; however, the decrease was not statistically significant (Figure 1b). The sleep quality score, assessed by the PSQI, decreased significantly over the 12-week intervention (mean ± SD: 0 weeks 10.1 ± 3.2, 12 weeks 8.2 ± 3.7; *p* = 0.023; Figure 1c). There was no significant change in the GSRS total score over the study period (Figure 1d), and there was no significant change in any of the 5 subscales of “Reflux”, “Diarrhea”, “Constipation”, “Abdominal pain”, or “Indigestion syndrome” (data not shown). The percentage of subjects diagnosed with IBS changed from 38.9% at 0 weeks to 16.7% at 12 weeks, and the state anxiety levels, as rated by the STAI, showed a downward trend (Table 2); however, neither change was statistically significant (*p* = 0.112 between Weeks 0–12 for both).

### 3.3. Changes in the Gut Microbiota

The changes in the gut microbiota are summarized in Appendix A. The majority of the bacterial targets were not significantly affected by the intervention. As expected, the counts of total lactobacilli increased significantly after the intervention began (*p* < 0.001 between Weeks 0–6, *p* = 0.019 between Weeks 0–12); and its subpopulation, Lacticaseibacillus (formerly referred to as the *Lactobacillus casei* subgroup), was observed to be a main component contributing to the increase.

### 3.4. Relationship between the Gut Microbiota and Reduction of Depressive Symptoms

We performed the responder analysis based on the extent of reduction of depressive symptoms. The subjects whose HAM-D21 score became half the baseline or less were designated as “responder” (*n* = 8), and the others as “non-responder” (*n* = 10) (Figure 2a). There were no significant differences between the two groups in the baseline HAM-D21 score, medication status, or diagnosis (MDD/BD) (Appendix A). At baseline (0 weeks), the responders showed significantly higher counts of Bifidobacterium and the Atopobium cluster than non-responders, although the differences tended to diminish over the intervention period (Figure 2b). The other bacterial targets, including total lactobacilli, did not show significant differences between the two groups at any time points (Appendix A).

We then calculated the area under the curve over the 12-week study (AUC_0–12_) of the gut microbiota, to evaluate the relationship between bacterial exposure over the time and reduction of depressive symptoms. The AUC_0–12_ of both Bifidobacterium and the Atopobium cluster had a significantly negative correlation with the HAM-D21 change (Figure 3). In the multiple linear regression analysis, the effect size of the AUC_0–12_ of the Atopobium cluster was statistically significant for the change in HAM-D21 score (estimate −0.49; 95% CI −0.82 to −0.16; *p* = 0.013), indicating that greater exposure to this bacterium was related to greater reduction of depressive symptoms. The AUC_0–12_ of Bifidobacterium was not significant for the HAM-D21 change (estimate −0.10; 95% CI −0.28 to 0.07; *p* = 0.256).

### 3.5. Changes in the Biological Markers of Intestinal Permeability and Inflammation

The L/M ratio and hs-CRP level were measured to evaluate intestinal permeability and inflammation in the subjects (Figure 4). The L/M ratio was 0.052 ± 0.033 (mean ± SD) at baseline, and remained at similar levels over the study period. No significant changes were observed in the hs-CRP level from its baseline value of 420.7 ± 469.9 ng/mL (mean ± SD). When these data were compared between responders and non-responders, there were no significant differences in either item at any time point (data not shown).

## 4. Discussion

We observed a significant reduction in depressive symptoms after the probiotic treatment, as evaluated by HAM-D21 (Figure 1a), which was accompanied by significant improvement in sleep quality and sleep disturbance, as assessed with the PSQI (Figure 1c), though the changes in the self-reported depressive symptoms, based on the BDI, were not statistically significant (Figure 1b). Insomnia is known as one of the core symptoms of depression and is estimated to occur in the majority of depressed patients [44]. The mean PSQI score of the patients in the present study was 10.1 at baseline, which exceeded the 5.5 cut-off point for insomnia [45]. The present results indicate that sleep quality improved with the reduction of depressive symptoms.

In an intervention study with a single-arm clinical trial design, several factors, including placebo effects, effects of medication and diet, and spontaneous remission, must be considered. We did not control diet, except for the administration of commercial probiotic products, and most of the patients continued taking antidepressants during the study. However, factors known to be associated with depression, specifically sleep quality and the onset of IBS [46], showed positive trends during the probiotic treatment (Figure 1c, Table 2). Although it is difficult to estimate the extent of the effect of LcS treatment alone, similar changes in the multiple measurements would indicate that the LcS treatment had positive effects on the improvement in depressive symptoms.

Several studies have evaluated the effects of LcS treatment in improving depressive mood [47], and in alleviating biological responses under stressful conditions [31,32]. Benton et al. conducted a randomized controlled trial in healthy subjects consisting mainly of aged adults, and LcS-treated subjects showed a significant improvement in depressive mood, compared to placebo-treated subjects, in a subgroup with a high depressive index at baseline [47]. In double-blind, randomized, placebo-controlled trials on healthy undergraduate medical students who were exposed to academic examination stress, LcS treatment alleviated stress-associated symptoms (elevation in salivary cortisol levels and sleep disturbance) [31,32]. It has been suggested that these actions might be mediated by a direct neural communication between the gut and brain through stimulation of the gastric branch of the vagal afferent by LcS [31,48]. The results of the present study have relevance to those studies and raise the possibility that LcS influences the brain functions to ameliorate depressive symptoms by modulating the gut–brain axis.

In the analysis of the gut microbiota, there was no significant change over the intervention period, except in the case of total lactobacilli, and the increase in total lactobacilli was attributed to its subpopulation, Lacticaseibacillus (Appendix A). The probiotic LcS strain contained in the study product taxonomically belongs to Lacticaseibacillus, and hence the changes in total lactobacilli and Lacticaseibacillus were attributable to intake of the study product. On the other hand, Bifidobacterium and the Atopobium cluster were found to be related to the magnitude of the intervention-associated improvement in depressive symptoms (Figure 2b), and the improvement was more pronounced when these bacteria were maintained at a high level (Figure 3). Interestingly, both bacteria taxonomically belong to the Actinobacteria phylum, which is one of the four major phyla of the gut microbiota. Actinobacteria are known to play a pivotal role in the development and maintenance of gut homeostasis, and their ability to produce a large amount of short chain fatty acids (SCFAs) is considered to contribute to the maintenance of the gut barrier [49]. The gut barrier plays an important role in the gut–brain interaction, and dysbiosis of the gut microbiota, along with the consequent increase in intestinal permeability, is associated with an up-regulation of systemic inflammation that may also involve the central nervous system [50]. SCFAs are also considered to play a key role in signal transduction through the gut–brain axis [10]. Long-term ingestion of LcS has been reported to increase the population levels of indigenous Bifidobacterium and the concentration of SCFAs [25,26]. No significant changes were here observed in the markers of intestinal permeability and inflammation (Figure 4), but it is possible that structural or functional changes occurred in Bifidobacterium and the Atopobium cluster as a result of LcS ingestion and modulated the gut–brain interaction. Several reports have indicated an association between Actinobacteria and psychiatric diseases [51]. Some showed a high abundance of Actinobacteria in MDD and BD patients [52,53], while we previously observed that MDD patients had lower Bifidobacterium counts than healthy controls [17]. Although the perspectives on the microbial communities in depression are inconsistent, the observed results indicate that Bifidobacterium and the Atopobium cluster of Actinobacteria may have synergetic effects with the LcS treatment.

Probiotics that confer such benefits are now called psychobiotics, which was originally defined as a live organism that, when ingested in adequate amounts, produces a health benefit in patients suffering from psychiatric illness [54]. The patients of this study took 8.0 × 10^10^ CFUs of live LcS daily for 12 weeks. The activity of LcS to stimulate the gastric branch of the vagal afferent was observed in a dose-dependent manner in rats [31], and daily intake of LcS with the amount of 1.0 × 10^11^ CFUs alleviated stress-associated symptoms in a human study of medical students under academic examination stress [31,32]. These results indicated that high-dose administration of LcS was one of the factors that confer a health benefit in patients with psychiatric symptoms. It has been reported that LcS reaches the intestine alive and contributes normalization of the gut microbiota [25,26]. These characteristics of LcS might have contributed to its synergetic effects with Bifidobacterium and the Atopobium cluster observed in this study. Recent reports showed that heat-killed lactobacilli strains conferred benefits to ameliorate chronic-stress-associated responses and sleep quality [55,56], and the mechanism of action due to the formulation of psychobiotics (live or dead) is a research topic for the future.

The study had several limitations in its study design, including no placebo control and a small sample size. Further investigations, involving a double-blind placebo-controlled trial, are necessary to determine to what extent the positive outcomes may be attributed to the LcS treatment. Another limitation lies in the fact that the majority of the participants were receiving psychotropic medications, although the medication remained the same over the course of the intervention.

## 5. Conclusions

We observed a significant improvement in depressive symptoms after LcS treatment. The improvement was associated with the gut microbiota, and more pronounced when Bifidobacterium and the Atopobium cluster were maintained at higher counts, suggesting that this strategy leads to improvement in depressive symptoms. Thus, interventions targeting these gut bacteria are considered to be candidates for future research, especially as it is possible that they may interact with LcS and exert synergetic effects on depressive symptoms.

## Figures and Tables

**Figure 1 microorganisms-09-01026-f001:**
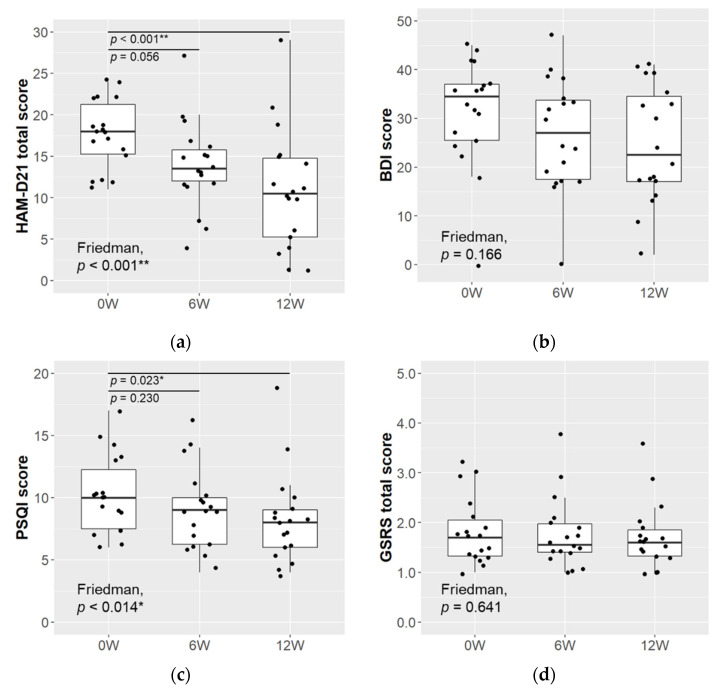
Changes in rating scales of depressive, sleep, and gastrointestinal symptoms. Depressive symptoms scores evaluated by (**a**) HAM-D21 and (**b**) BDI; sleep quality and sleep disturbance rated by (**c**) PSQI; GI symptoms score evaluated by (**d**) GSRS (*n* = 18). The Friedman test was used for the multiple comparisons between time points, and when significance was observed, the Nemenyi test was conducted post hoc for pairwise comparisons with the baseline data (* *p* < 0.050, ** *p* < 0.010).

**Figure 2 microorganisms-09-01026-f002:**
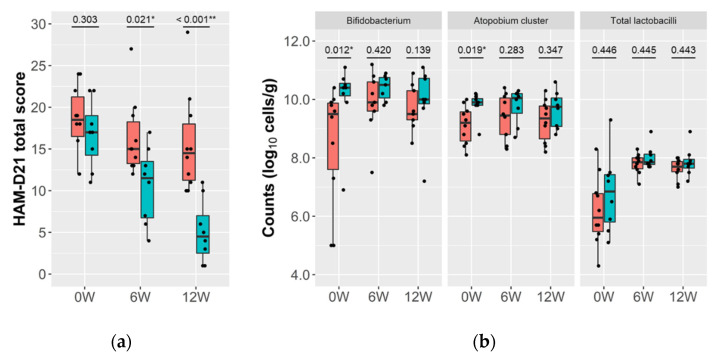
Responder analysis based on the change in HAM-D21 total score. The responder subgroups were designated based on the percent change in the HAM-D21 total score over the 12-week intervention period. Subjects whose percent change was −50.0% or less were designated as “responder” (*n* = 8; blue bars), with the remaining participants designated as “non-responder” (*n* = 10; red bars) (**a**). Changes in the counts of Bifidobacterium, the Atopobium cluster, and total lactobacilli, are shown for the two groups (**b**). Wilcoxon’s rank sum test was used to compare the data between the two groups (* *p* < 0.050, ** *p* < 0.010), and the *p*-value is shown for each pair.

**Figure 3 microorganisms-09-01026-f003:**
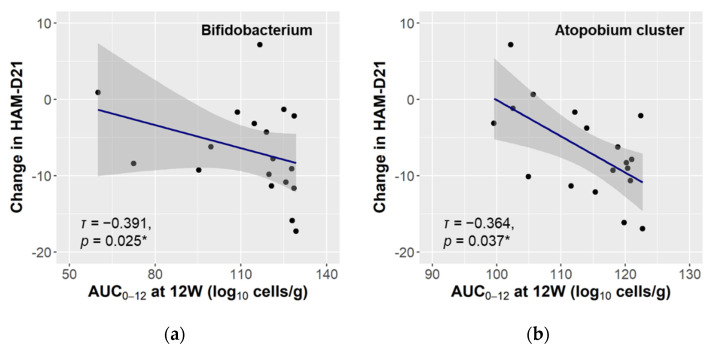
Correlation between the gut microbiota and change in the severity of depressive symptoms. The 12-week AUCs of the bacterial counts of Bifidobacterium (**a**) and the Atopobium cluster (**b**) were calculated, and their statistical association with the change in the HAM-D21 score over the same period was evaluated (*n* = 18). The linear regression curve with 95% CI is plotted (grey area), and Kendall’s rank correlation coefficient (*τ*) and *p*-value (* *p* < 0.050) are shown.

**Figure 4 microorganisms-09-01026-f004:**
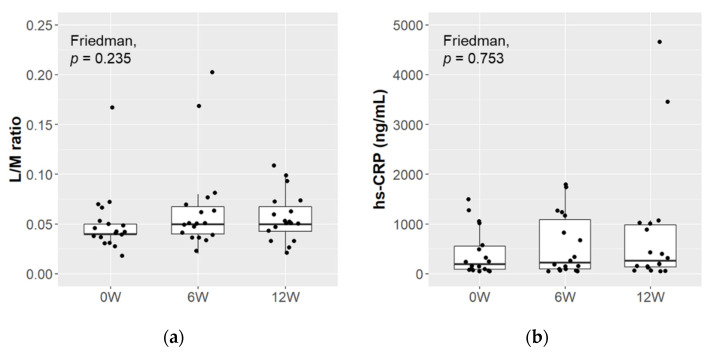
Changes in the markers of intestinal permeability and inflammation. The L/M ratio (**a**) and hs-CRP level (**b**) were measured to evaluate intestinal permeability and inflammation, respectively (*n* = 18). The Friedman test was used for the multiple comparisons between time points.

**Table 1 microorganisms-09-01026-t001:** Demographic and clinical characteristics of the subjects at baseline.

	Mean ± SD	95% CI	Median (min, max)
Age, years	40.6 ± 11.4	34.9–46.3	43 (21, 55)
Education, years	15.0 ± 2.2	13.9–16.1	15.5 (9, 18)
BMI, kg/m^2^	21.80 ± 4.22	19.70–23.90	20.6 (16.5, 34.8)
HAM-D21 total score	17.7 ± 4.1	15.6–19.7	18 (11, 24)
BDI score	31.5 ± 11.0	26.0–37.0	35 (0, 45)

**Table 2 microorganisms-09-01026-t002:** Changes in IBS diagnosis and state anxiety levels rated by the STAI.

	Number (%)			*p*-Value ^b^	
	0W	6W	12W	W0–6	W0–12
IBS diagnosis					
Normal	5 (27.8)	5 (27.8)	7 (38.9)	NA	0.112
FBD	6 (33.3)	10 (55.6)	8 (44.4)		
IBS	7 (38.9)	3 (16.7)	3 (16.7)		
STAI state, level ^a^					
Normal	1 (5.6)	4 (22.2)	4 (22.2)	NA	0.112
High	4 (22.2)	3 (16.7)	6 (33.3)		
Very high	13 (72.2)	11 (61.1)	8 (44.4)		

^a^ The state anxiety level of the STAI was defined as follows: “very low” (male: 20 to 22, female: 20 to 21), “low” (23 to 31, 22 to 30), “normal” (32 to 40, 31 to 41), “high” (41 to 49, 42 to 50), and “very high” (50 to 80, 51 to 80). No subjects scored “very low” or “low”. ^b^ McNemar Chi-squared test was used to compare paired proportions between the time points (NA, not applicable).

## Data Availability

The data are not available due to the nature of this research. Participants of this study did not agree for their data to be shared publicly.

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
