# Peer review of "Effect of Lacticaseibacillus paracasei Strain Shirota on Improvement in Depressive Symptoms, and Its Association with Abundance of Actinobacteria in Gut Microbiota"

_microorganisms, 2021, doi:10.3390/microorganisms9051026_

Round 1

Reviewer 1 Report

Introduction should be revised. Please avoid using “we”.

Introduction. Please provide the significance of the present study in the text.

Although it is not necessary, in my opinion, the use of separate section of conclusions would be better.

The work provides its limitations and possible future studies. It is well prepared and in my opinion is a great contribution in the field.

Author Response

Comment 1)

Introduction should be revised. Please avoid using “we”.

Response to Comment 1)

The reviewer recommended to revise the Introduction section to avoid using the pronoun “we”, but we could not see the significance of this suggestion. We have presumed that the comment was not on the Introduction section but on the Abstract section, and have amended the Abstract section (lines 32 – 34 in the revised version). We have kept the first sentence as it was so as to specify who performed the previous study that became a rationale for the current research (lines 20 – 21 in the revised version).

Comment 2)

Introduction. Please provide the significance of the present study in the text.

Response to Comment 2)

According to the reviewer’s suggestion, we have added the sentence below in the Introduction section to define the significance of the present study: “The study will provide key new findings that will help elucidate the mechanisms of the psychological effects of probiotics.” (lines 87 – 88 in the revised version).

Comment 3)

Although it is not necessary, in my opinion, the use of separate section of conclusions would be better.

Response to Comment 3)

According to the reviewer’s suggestion, we have added the Conclusion section and have moved the last paragraph in the Discussion section to there (lines 349 – 356 in the revised version).

Comment 4)

The work provides its limitations and possible future studies. It is well prepared and in my opinion is a great contribution in the field.

Response to Comment 4)

We appreciate the very positive comments on our manuscript.

Reviewer 2 Report

In the present MS, Machiko Otaka al, presented an interesting study about the effect of Lacticaseibacillus paracasei strain Shirota on recovery from depressive symptoms.

Regarding the title, Effect of Lacticaseibacillus paracasei strain Shirota on recovery from depressive symptoms….I consider that should be modified. The results obtained could be interpreted as an improvement but not a recovery.

The kits used for biological markers for intestinal permeability and inflammation are not clear, what type of kits they did use? Please, indicate the protocol or kits used.

The score for Hamilton is really low for this type of study.  Authors should explain the reason for establishing a score of 11 for Hamilton Scale.

Have they found strong differences by different score level: 10-13; 14-17?

Statistic for Table 3 is not clear. According to the legend:  Wilcoxon’s rank sum test was used for the inter-group comparison at each time point (*, p < 0.050). Means that 10.4 is different to 9.5 with a significant of p<0.05?????

Statistic is confusing, Regarding the proportion of Prevotella (tabe S1), this bacteria has been associated to MDD but in this case seems to be a reduction, no significative?¿ ( 6.0 (2.5, 8.0) vs 2.5 (2.5, 6.8),  2.5 (2.5, 7.0). Authors should do mention to this finding

Author Response

In the present MS. Machiko Otaka et al. presented an interesting study about the effect of Lacticaseibacillus paracasei strain Shirota on recovery from depressive symptoms.

Comment 1)

Regarding the title, Effect of Lacticaseibacillus paracasei strain Shirota on recovery from depressive symptoms….I consider that should be modified. The results obtained could be interpreted as an improvement but not a recovery.

Response to Comment 1)

We agree with the reviewer’s suggestion, and have modified the title as below: “Effect of Lacticaseibacillus paracasei strain Shirota on improvement in depressive symptoms, and its association with abundance of Actinobacteria in gut microbiota” (lines 2 – 4 in the revised version). We also have changed the corresponding term in the main text from “recovery from” to “reduction of” or “improvement in” (lines 29, 224, 225, 236, 241, 276, 285, 305, 306, 350, and 351 in the revised version).

Comment 2)

The kits used for biological markers for intestinal permeability and inflammation are not clear, what type of kits they did use? Please, indicate the protocol or kits used.

Response to Comment 2)

We used the kit named Intestinal Permeability Assessment (Genova Diagnostics; Asheville, NC, US) for intestinal permeability test and followed the manufacturer’s instructions. Regarding the measurement of hs-CRP, we outsourced this to the company for clinical laboratory services. We have mentioned those in the text (lines 144 – 154 in the revised version).

Comment 3)

The score for Hamilton is really low for this type of study. Authors should explain the reason for establishing a score of 11 for Hamilton Scale.

Response to Comment 3)

Thank you for the important point. We set the inclusion criterion regarding the Hamilton scale at a score of 11 or more to see the effect of LcS in patients with a wide range of severity including milder forms of the illness. We have added the descriptions at the corresponding part (lines 95 – 99 in the revised version).

Comment 4)

Have they found strong differences by different score level: 10-13; 14-17?

Response to Comment 4)

Thank you for raising an interesting point. However, since the number of participants was small (n = 18), we have refrained from performing such a stratified analysis.

Comment 5)

Statistic for Table 3 is not clear. According to the legend: Wilcoxon’s rank sum test was used for the inter-group comparison at each time point (*, p < 0.050). Means that 10.4 is different to 9.5 with a significant of p<0.05?????

Response to Comment 5)

The reviewer’s understanding is correct. Wilcoxon’s rank sum test was used to compare the counts between the subgroups, and the bifidobacterial counts of “responder” (median 10.4) were higher than those of “non-responder” (median 9.5) with a significance of p<0.05. We have added descriptions for clarify in the foot note of Table S3.

Comment 6)

Statistic is confusing, Regarding the proportion of Prevotella (Table S1), this bacteria has been associated to MDD but in this case seems to be a reduction, no significative? (6.0 (2.5, 8.0) vs 2.5 (2.5, 6.8), 2.5 (2.5, 7.0). Authors should do mention to this finding

Response to Comment 6)

The data was expressed as median (min, max) in the table. As mentioned in the Materials and Methods section, bacterial counts of “not detected” samples were regarded as representing half the detection limit in the statistical analysis (the detection limit of Prevotella was 5.0). The detection rate of this bacterial genera was 11/18 at 0 weeks, 8/18 at 6 weeks, and 7/18 at 12 weeks. The median of 0 weeks was an actual value (6.0), while that of 6 and 12 weeks was an imputed value (2.5). They might look reduced after the probiotic treatment, but in practice, the change was not statistically significant. We do not think this is such a key finding, and thus have decided not to give a topic on this result.

Reviewer 3 Report

I have read with great interest the paper written by Otaka et al. on the effect of Lactobacilli on depressive symptoms.

Introduction:

Lines 38-42: little to none general information is provided for gut microbiota, the authors should cite this recent paper that covers most of gut-microbiota basics knowledge:DOI: 10.1152/ajpgi.00161.2019

Lines 43-45: in terms of gut brain axis, the authors should provide more information and cite this recent paper on ENS/gut-brain axis: DOI:10.3390/jcm9113705

Statistical analysis: was p-value two-tailed? please, specify.

In the discussion section the authors should point out the current challenges in probiotics/prebiotics treatment (are they effective in reaching the intestine? what formulation? how much does their effect last?) and how this can affect the research.

Author Response

I have read with great interest the paper written by Otaka et al. on the effect of Lactobacilli on depressive symptoms.

Comment 1)

Introduction:

Lines 38-42: little to none general information is provided for gut microbiota, the authors should cite this recent paper that covers most of gut-microbiota basics knowledge: DOI: 10.1152/ajpgi.00161.2019

Response to Comment 1)

We have cited the paper that the reviewer recommended in the Introduction section (line 42 in the revised version).

Comment 2)

Lines 43-45: in terms of gut brain axis, the authors should provide more information and cite this recent paper on ENS/gut-brain axis: DOI: 10.3390/jcm9113705

Response to Comment 2)

According to the reviewer’s suggestion, in the Introduction section, we have added descriptions regarding the gut–brain axis and have cited the paper that the reviewer recommended (lines 44 – 48 and 56 in the revised version).

Comment 3)

Statistical analysis: was p-value two-tailed? please, specify.

Response to Comment 3)

Two-tailed p-values were presented in the manuscript. We have added descriptions “All statistical tests were performed in a two-tailed test at the 0.050 level of significance.” in the Materials and Methods section (lines 172 – 173 in the revised version).

Comment 4)

In the discussion section the authors should point out the current challenges in probiotics/prebiotics treatment (are they effective in reaching the intestine? what formulation? how much does their effect last?) and how this can affect the research.

Response to Comment 4)

According to the reviewer’s suggestion, we have added a paragraph in the Discussion section to discuss the formulation of probiotics: daily dose and live/dead (lines 328 – 342 in the revised version).